# Sexually Dimorphic Behavioral Profile in a Transgenic Model Enabling Targeted Recombination in Active Neurons in Response to Ketamine and (2R,6R)-Hydroxynorketamine Administration

**DOI:** 10.3390/ijms21062142

**Published:** 2020-03-20

**Authors:** David P. Herzog, Ratnadevi M. Mellema, Floortje Remmers, Beat Lutz, Marianne B. Müller, Giulia Treccani

**Affiliations:** 1Laboratory of Translational Psychiatry and Focus Program Translational Neurosciences, Johannes Gutenberg University Medical Center Mainz, 55128 Mainz, Germany; daherzog@uni-mainz.de (D.P.H.); ratna_mellema@msn.com (R.M.M.); marianne.mueller@uni-mainz.de (M.B.M.); 2Institute of Physiological Chemistry, Johannes Gutenberg University Medical Center Mainz, 55128 Mainz, Germany; remmersf@uni-mainz.de (F.R.); beat.lutz@uni-mainz.de (B.L.); 3Leibniz Institute for Resilience Research, 55131 Mainz, Germany; 4Translational Neuropsychiatry Unit, Department of Clinical Medicine, Aarhus University, 8000 Aarhus, Denmark

**Keywords:** ketamine, hydroxynorketamine, antidepressant, rapid-acting, sex difference, BDNF, activated neurons, behavior

## Abstract

Background: Rapid-acting antidepressants ketamine and (2R,6R)-hydroxynorketamine ((2R,6R)-HNK) have overcome some of the major limitations of classical antidepressants. However, little is known about sex-specific differences in the behavioral and molecular effects of ketamine and (2R,6R)-HNK in rodents. Methods: We treated mice with an intraperitoneal injection of either saline, ketamine (30 mg kg^−1^) or (2R,6R)-HNK (10 mg kg^−1^). We performed a comprehensive behavioral test battery to characterize the Arc-CreERT2 × CAG-Sun1/sfGFP mouse line which enables targeted recombination in active populations. We performed a molecular study in Arc-CreERT2 × CAG-Sun1/sfGFP female mice using both immunohistochemistry and in situ hybridization. Results: Arc-CreERT2 × CAG-Sun1/sfGFP mice showed sex differences in sociability and anxiety tests. Moreover, ketamine and (2R,6R)-HNK had opposite effects in the forced swim test (FST) depending on gender. In addition, in male mice, ketamine-treated animals were less immobile compared to (2R,6R)-HNK, thus showing a different profile of the two drugs in the FST. At the molecular level we identified *Bdnf* mRNA level to be increased after ketamine treatment in female mice. Conclusion: Arc-CreERT2 × CAG-Sun1/sfGFP mice showed sex differences in social and anxiety behavior and a different pattern between ketamine and (2R,6R)-HNK in the FST in male and female mice. At the molecular level, female mice treated with ketamine showed an increase of *Bdnf* mRNA level, as previously observed in male mice.

## 1. Introduction

Major depressive disorder (MDD) poses a serious threat on modern societies [1] and novel treatment approaches besides classical antidepressants are needed to advance treatment of this devastating illness. In recent decades, the emergence of rapid-acting antidepressants in preclinical and clinical studies have overcome some of the limitations of MDD therapy: the effect latency [2] and the high rate of non-response [3] to most classical antidepressants.

The rapid-acting antidepressant ketamine has been repeatedly reported to be effective in both animal models of MDD [3] and patients [4]: in contrast to classical antidepressants, ketamine exerted its effects within hours [5], lasting for several days [5], and was even found effective in patients with treatment-resistant MDD [5]. However, ketamine treatment comes with some relevant side effects like nausea, sedation, and the risk of addiction, which so far prevented its wide-spread use [6]. So far, ketamine administration has had to be via parenteral administration, further impairing its applicability in the clinic. However, just recently a new ketamine nasal spray in combination with an oral antidepressant has been approved by the FDA for treatment-resistant depression, hence providing a new available formulation [7]. To avoid the ketamine side-effects, researchers also looked into the various ketamine metabolites, lacking these side-effects. Zanos and colleagues recently showed that (2R,6R)-hydroxynorketamine ((2R,6R)-HNK) is a ketamine metabolite with similar antidepressant-like efficacy in rodents, but without the ketamine-related side effects [8].

The antidepressant ketamine in mice affects several behavioral domains. It was found to be anxiolytic [9] as well as to reduce the immobility time in the forced swim test (FST), a commonly used test for depressive-like behavior [10]. However, long-term administration of ketamine seemed to impair cognitive function in mice [11]. Many behavioral studies observed a significant difference in the behavioral effects of ketamine based on gender. Indeed, female mice were reported to be more susceptible to the effects of ketamine: they respond to lower doses of ketamine [12] and exhibit more pronounced acute and long-term antidepressant-like effects [13]. These differences might be explained by sex-specific alterations of hormones and metabolism: estrogen receptor agonists lowered the dose required to produce an antidepressant-like effect in female mice [12] and female rodents developed higher plasma and brain ketamine concentrations, followed by a slower elimination of the substance from the organism [14]. Studies addressing the role of gender in (2R,6R)-HNK treatment are limited. Published work so far did not detect a role of gender in the behavioral response to (2R,6R)-HNK administration. Recently, we summarized the evidence of the role of sex in the effects of ketamine and (2R,6R)-HNK [15].

The molecular mechanisms of action of ketamine and (2R,6R)-HNK are only partly understood: there are key neurobiological mechanisms that are well-known factors of MDD pathology and therapy, which were found to play a role in the rapid-acting effects of ketamine. For example, a single injection of ketamine activated the mammalian target of rapamycin (mTOR) and this enhanced synaptogenesis in the medial prefrontal cortex (mPFC) [16], reversing the stress-induced synaptic deficits in mice [17]. Ketamine administration led to the fast production of brain neurotrophic factor (BDNF) [18], an important mediator of MDD pathology and treatment [19], by inducing glutamate bursts and activating postsynaptic α-amino-3-hydroxy-5-methyl-4-isoxazolepropionic acid receptor (AMPA) receptors [20]. Similar to ketamine, in vitro [21] and in vivo [22] experiments showed that (2R,6R)-HNK induced structural and synaptic plasticity. A similar involvement of the mTOR pathway and BDNF in the mechanisms of action of (2R,6R)-HNK is likely. Just recently, Fukumoto et al. showed that BDNF signaling is required for the antidepressant-like effects of (2R,6R)-HNK [23].

The behavioral and molecular effects of ketamine and (2R,6R)-HNK were in the focus of the present study. We analyzed these effects using a transgenic mouse line capable of a targeted recombination in active populations (TRAP), the Arc-CreERT2 × CAG-Sun1/sfGFP [24]. This mouse line allows the permanent labeling of activated neurons by combining a green fluorescent protein (GFP) conditional reporter line, and a tamoxifen (TM)-dependent Cre-line wherein the expression of Cre is under the control of an immediate early gene promoter, in this case under the activity-regulated cytoskeleton-associated protein (*Arc*) promoter [25]. Ketamine is known to enhance Arc expression [25] and hence in this line a GFP signal is produced in any neuron with a rise in Arc expression [24]. So far, a comprehensive and sex-specific behavioral characterization of the effects of ketamine and HNK on the Arc-CreERT2 × CAG-Sun1/sfGFP mouse line is lacking. First, we wanted to fill this gap of knowledge by applying a condensed behavioral battery to assess female and male Arc-CreERT2 × CAG-Sun1/sfGFP mice in several behavioral domains. In a second step, we performed a molecular analysis elucidating the effects of ketamine and (2R,6R)-HNK on *Bdnf* expression using Arc-CreERT2 × CAG-Sun1/sfGFP female mice.

## 2. Results

### 2.1. Behavioral Characterization

Arc-CreER^T2^ × CAG-Sun1/sfGFP mice were subjected to a test battery that enabled characterization of multiple behavioral domains in a limited amount of time [26]. These experiments were designed in order to characterize the behavioral phenotype of female and male mice in the absence of (tamoxifen induced) recombination and yet to understand which are the behavior changes in response to ketamine and (2R,6R)-HNK treatment.

#### 2.1.1. Ketamine and (2R,6R)-HNK Do not Affect Spatial (Working) or Episodic Memory

Statistical analysis of the spontaneous alternations, spatial object recognition task (SORT) and novel object recognition (NORT) task demonstrated no effect of treatment (Figure 1A–C, Table 1), suggesting that ketamine and (2R,6R)-HNK did not affect hippocampal-dependent memory performance 24 h after administration. In the spontaneous alternation test all treatment groups revealed around 60% spontaneous alternation performance (SAP) behavior, indicating intact working memory (Figure 1A, Table 1 and Table 2 for detailed statistics). In the SORT test, only female mice showed significant spatial object recognition above chance level (50% cut off), whereas males did not (Figure 1B, Table 1 and Table 2). In the NORT, saline-treated, but neither ketamine nor (2R,6R)-HNK treated females showed significant novel object recognition above chance level (Figure 1C, Table 1 and Table 2).

#### 2.1.2. Sex, but Not Treatment, Affected Social Behavior and Anxiety 

Other behavioral domains that were examined in the behavioral battery comprised locomotion, sociability preference, and anxiety-like behavior. Statistical analysis of performance in the open field (OF) revealed that ketamine and (2R,6R)-HNK did not affect the total distance mice travelled (Figure 1D, Table 1). All treatment groups travelled an average distance of 5.5 m in the OF, independent of sex. Statistical analysis of the sociability test showed that ketamine and (2R,6R)-HNK did not affect sociability (Table 1, Figure 1E). All treatment groups showed significant preference for the social object compared to chance level (see Table 1 and Table 2 for statistics), suggesting intact social behavior. In contrast, sex had a significant main effect on sociability (see Table 1 for statistics). Likewise, anxiety-like behavior, measured as avoidance of the lit compartment of the light-dark (LD) box (Figure 1F) was significantly affected by sex (Table 1). Importantly, statistical analysis showed that anxiety-like behavior tested in the LD box was not affected by ketamine nor (2R,6R)-HNK (Table 1). Neither treatment nor sex affected olfaction in the bedding preference test (Figure A1 and Table 1).

#### 2.1.3. Sex Influenced the Antidepressant-Like Effects of Ketamine and (2R,6R)-HNK in the Forced Swim Test (FST)

To assess the antidepressant-like effect of ketamine and (2R,6R)-HNK in the Arc-CreER^T2^ × CAG-Sun1/sfGFP mouse line, male and female mice were subjected to the forced swim test 24 h after a single injection with either saline, ketamine, or (2R,6R)-HNK. Statistical analysis determined a significant interaction: the effects of saline, ketamine, or (2R,6R)-HNK were opposite in male and female mice (Figure 2) with males more immobile after saline and (2R,6R)-HNK treatment and less immobile after ketamine treatment, while female showed the opposite in each condition. We also detected a statistical trend for sex difference. Post-test analysis also revealed that ketamine treated male mice were less immobile than those treated with (2R,6R)-HNK, hence suggesting that ketamine and (2R,6R)-HNK revealed different effects in the FST (Figure 2).

### 2.2. Molecular Characterization

A key component in the mechanism of action of ketamine is the activation of neuroplasticity mechanism and among those, the increase of BDNF [18]. Surprisingly, BDNF increase has been mainly reported in male mice. Hence, it is of interest to confirm that a similar mechanism is taking place in female mice. By taking advantage of the use of the Arc-CreER^T2^ × CAG-Sun1/sfGFP mice, we identified molecular changes only occurring in activated neurons, wherein tamoxifen-induced targeted recombination took place in response to stimuli, in our case ketamine, (2R,6R)-HNK, or saline treatment. 

#### 2.2.1. Activated Nuclei Upon Ketamine and (2R,6R)-HNK Administration

In the Arc-CreERT2 × CAG-Sun1/sfGFP mice, TM administration causes active (Arc-expressing and therefore) CreERT2-expressing cells to undergo Cre-mediated recombination (to be ‘‘TRAPed’’), resulting in permanent expression of the effector gene (GFP; Figure 3A–C). Nonactive cells do not express CreERT2 and do not undergo recombination, even in the presence of TM. We therefore quantified the mean intensity of GFP labelled activated nuclei of female mice. Statistical analysis did not show any significant differences between treatment groups in neither the dentate gyrus (DG) nor the Cornu Ammonis region 3 (CA3) (Figure 3D,E). However, a strong trend towards significance was evident in the CA3 region of the hippocampus.

#### 2.2.2. Ketamine Increase BDNF Intensity in DG and CA3

In order to confirm in females that ketamine and (2R,6R)-HNK increase *Bdnf* mRNa levels, we performed in situ hybridization (Figure 4A,B). The fluorescence mean intensity of *Bdnf* mRNA expression was measured for each condition. Statistical analysis demonstrated a significant difference in *Bdnf* level between groups in both DG (Figure 4C) and CA3 (Figure 4D). Bonferroni post-hoc test revealed a significant difference in *Bdnf* intensity between ketamine and saline treated females in both DG and CA3. Interestingly, in none of the hippocampal regions a significant difference was found between saline and (2R,6R)-HNK or ketamine and (2R,6R)-HNK (Figure 4).

## 3. Discussion

Sex differences in neuropsychopharmacology have been overlooked for several years. Recently, rodent studies revealed that female and male are very diverse in several aspects. They showed different behavioral phenotype in multiple tests and they showed differences in pharmacokinetics and pharmacodynamic of several drugs. A very dramatic consequence for these differences is that a drug mainly developed and tested in male subjects, might show severe side effects in female subjects or not even show efficacy [27].

In the search for better antidepressant drugs, ketamine and its metabolite (2R,6R)-HNK represent a realistic new alternative [28]. However, the differences driven by sex in the antidepressant property of the two drugs are as yet fragmented. In this study we performed a baseline characterization on the sex differences of the Arc-CreER^T2^ × CAG-Sun1/sfGFP mice treated with ketamine and (2R,6R)-HNK. Using a test battery previously established by our group [26], we found that both ketamine and (2R,6R)-HNK did not affect memory, anxiety-like behavior, sociability, and locomotion 24 h after administration but that sex significantly influenced the mouse performance in the sociability and anxiety tests [8,29,30,31,32]. That neither ketamine nor (2R,6R)-HNK influenced memory, sociability, nor locomotion is in line with previous findings. However contrary to our results ketamine has been previously reported to act as an anxiolytic [9,33]. An explanation for this divergence could be found in the dose at which ketamine was used. Previous research suggested that the anxiolytic effect of ketamine is dose dependent, with low doses leading to anxiolytic effects, but high doses increasing anxiety symptoms [34]. With a dose of 10 mg kg^-1^ the anxiolytic effect was detectable [33], thus the dose used in the current study may have been too high to elicit anxiolytic effects. On the other hand, the evidence for an anxiolytic effect of ketamine seems to be rare and ambiguous, as studies found anxiolytic effects [9,33], but others found anxiogenic effects [35]. 

Interestingly, in our study we found a significant sex difference in the sociability test and LD box test showing that these behaviors might be affected by sex.

As a classical tool to study antidepressant-like effects of drugs, we subjected male and female mice to the FST. We found a sex × treatment interaction. Indeed, in the FST ketamine and (2R,6R)-HNK had opposite effects in female and male mice. Moreover, we showed that ketamine and (2R,6R)-HNK treated male mice differed in the FST, with ketamine treated mice being less immobile compared to (2R,6R)-HNK treated mice. However, in our study we could not find a statistical difference between ketamine treated male mice and saline treated ones. The absence of this difference may have been caused by the absence of a stressor in our experimental procedure [10]. It was shown that stressed animals administered with ketamine at the same dose used in this study exhibited antidepressant-like effects, whereas unstressed animals did not. This highlights the importance of the implementation of a stressor beforehand for ketamine and (2R,6R)-HNK to elicit their antidepressant-like effect. Moreover, in the rodent literature there is no consensus yet on the antidepressant-like effects of ketamine. Additionally, these effects seem to be dependent on the dose, species and strain, test, stressor, and the sex of the experimenter as highlighted in a recent review [36]. Therefore, we once more highlight that the interpretation of our findings must remain limited to the sex of the tested animals for each test/measurement and to the specific doses that we applied in our studies.

Since we demonstrated an opposite effect in female and male mice in the FST, we wanted to understand whether at molecular level females would show a different pattern compared to what is reported in the literature for male mice. Therefore, we focused on BDNF which is widely considered as a main downstream mediator in the effect of ketamine [18] 

By using the elegant design of the Arc-CreER^T2^ × CAG-Sun1/sfGFP mice, we were able to, in the presence of TM, permanently label the neurons that upon activation underwent Cre-dependent recombination. Firstly, we analyzed the GFP intensity level in all experimental groups and hypothesized an increase of GFP intensity dependent on the activation on the Arc promoter (see Figure 3A). Previously, studies indeed reported that ketamine induced Arc expression [25,37]. Although no significant increases in GFP intensity were found in either the DG or the CA3, we found a strong trend towards significance in the CA3 region of the hippocampus. However, a recent study using the same line lacked in finding a significant increase in the number of activated cells after ketamine treatment in a fear conditioning experiment [38], but the authors were able to detect an increase in the colocalization between activated neurons and c-fos (the protein of interest).

Surprisingly, in our study, we found a significant increase in *Bdnf* mRNA expression in the DG and CA3 region of the dorsal hippocampus of females injected with ketamine compared to control. These findings suggested a mismatch between behavioral and molecular findings. Ketamine revealed different effects in the FST in male and female but led to an increase in *Bdnf* level in females (Figure 4) as previously reported for males [18]. Strikingly, no significant increase in *Bdnf* expression was found in the dorsal hippocampus of females injected with (2R,6R)-HNK. These findings are supported by a recent study that identified the medial prefrontal cortex as the key region for the actions of (2R,6R)-HNK [23]. The present results, together with these previous findings, suggest that (2R,6R)-HNK acts through a different neuroanatomical network than ketamine in female mice. Unfortunately, as the main limitation of this study, we were not able to stain BDNF protein in the GFP positive activated neurons. BDNF antibodies show indeed discontinuity among batches and lack of reproducibility in staining. Moreover, we performed the molecular study only in female mice, thus a comparison with male mice is only possible based on the literature. Therefore, further studies on a different cohort of animals might establish an increase of BDNF in the subset of GFP positive activated neurons both in male and female mice. 

## 4. Materials and Methods 

### 4.1. Animals

Heterozygous Arc-CreER^T2^ mice (Jax #022357) [24,37,39] and homozygous R26-CAG-LSL-Sun1/sfGFP-myc mice (Jax #021039) [40] were obtained from The Jackson Laboratory (USA) and were bred and crossed in-house to generate heterozygous Arc-CreER^T2^ × CAG-Sun1/sfGFP mice. Both male and female offspring heterozygous for both the reporter and CreER^T2^ gene were used in the experiments. At the age of 6 weeks, mice left the breeding area and entered our mouse behavioral unit. Mice were allowed to habituate to the new environment for at least one week. Mice were single-housed one week prior to the experiment. After entering our mouse behavioral unit, male and female mice were always housed in separate rooms.

Mice were housed in a room with 24 ± 1 °C and 40% humidity and maintained on a 12:12 h light/dark cycle with lights on at 7 am during the whole experiment. All animals were provided with food and water ad libitum. All experiments were carried out in accordance with the European Community’s Council Directive of 22 September 2010 (2010/63EU) and approved by the local authorities (Animal Protection Committee of the State Government, Landesuntersuchungsamt Rheinland-Pfalz, Koblenz, Germany).

### 4.2. Compounds

Tamoxifen (Cat#T5648, Sigma, St. Louis, MO, USA) was dissolved in 99.8% ethanol to reach a concentration of 100 mg mL^−1^. The tamoxifen/ethanol mixture was added to corn oil (Cat#C8267, Sigma, St. Louis, MO, USA) to reach a final concentration of 10 mg mL^−1^ and incubated for 15 min at 37 °C. To induce Cre recombination, the tamoxifen mixture was administered by a single intraperitoneal injection (150 mg kg^−1^) followed by a single intraperitoneal injection with one of the compounds, saline or ketamine or (2R,6R)-HNK. Tamoxifen was injected to make active CreERT2-expressing cells undergo Cre-mediated recombination, resulting in permanent expression of GFP. Intensity of the GFP protein was measured after immunohistochemistry (see Section 4.5). Racemic ketamine (Inresa Arzneimittel GmbH, Freiburg, Germany) and (2R,6R)-HNK (Tocris Bioscence, Bristol, UK) were dissolved in NaCl 0.9% (Saline, Braun, Melsungen, Germany) following manufactural descriptions. A single intraperitoneal injection of either ketamine (30 mg kg^−1^) [41], (2R,6R)-HNK (10 mg kg^−1^) [8], or saline (100 µL) was administered 24 h prior to behavioral testing. 

### 4.3. Behavioral Procedure

Mice were handled for 2 min per day on the five days preceding an injection with one of the compounds. Mice were randomly assigned to a treatment group. Twenty-four hours prior to the behavioral battery or forced swim test (FST), mice obtained a single intraperitoneal injection with saline, ketamine, or (2R,6R)-HNK. Test battery was performed in a sound attenuated environment. Male and female mice were tested in separate rooms. Behavioral testing was conducted in the light phase (9–12 a.m.) under dimmed light conditions (39 ± 2 lux), except during the light dark box (600 ± 2 lux). Animals were randomized and experimenters were blinded for the treatment conditions during testing, scoring, and analyses.

#### 4.3.1. Behavioral Battery

The behavioral battery enables a brief (3 h) and condensed testing of multiple important behavioral domains. The present behavioral battery was slightly modified from the behavioral battery described by Jene and colleagues [26]. Additional to the spontaneous alternations, spatial object recognition task (SORT), novel object recognition task (NORT), sociability test, light-dark box, and bedding preference task, the present behavioral battery included an open field (OF) to measure locomotor activity for 10 min which was conducted in an OF arena (45 × 45 × 41 cm) prior to the NORT (Figure 5). During inter-test intervals (i.t.i.), lasting 15 min, set-ups were cleaned with 5% ethanol.

#### 4.3.2. Forced Swim Test (FST)

Depressive-like behavior was assessed in the FST in a separate batch of animals. Each animal was placed in a glass cylinder (height: 24 cm; diameter: 13 cm) filled with water (depth: 15 cm; temperature: 21 ± 1 °C) for 5 min while behavior was video recorded from the side. Videos were manually scored on swimming, climbing, and floating behavior. Floating is represented as immobility time (s). Swimming and climbing behavior are scored together as mobility time (s). Water temperature was measured and adjusted following each test.

### 4.4. Tissue Collection

Drug-naïve female mice received two intraperitoneal injections with tamoxifen and with either saline, ketamine, or (2R,6R)-HNK and were maintained in an isolated, sound-attenuated room (temperature: 22 °C). Three days following the injections, mice were anesthetized with pentobarbital (100 mg kg^−1^ bodyweight) and buprenorphine (0.1 mg kg^−1^ bodyweight) and manually perfused with 10mL cold PBS1x (Thermofisher Scientific, USA) followed by 10 mL cold 4% paraformaldehyde (PFA, Santa Cruz Biotechnology, USA). Brains were dissected and post-fixed in 4% PFA overnight at 4 °C, treated with 30% sucrose for two days and stored at −80°C until further processing. 

### 4.5. Immunohistochemistry

Female brains were sectioned at 30 µm thickness in coronal plane using a Kryostat HM 560 Microm and stored in cryoprotectant (20% glycerol (87%), 30% ethylene glycerol, 50% PBS1x) until use. Immunohistochemistry was performed on 30µm dorsal hippocampal sections of the female brain. Sections were rinsed three times with PBS1x for 5 min to remove cryoprotectant, followed by a wash with PBS1x/0.2%Triton-X100(TX) for 5 min. Sections were blocked with 4% goat serum for 15 min and incubated with a primary antibody against GFP (Aveslabs cat#GFP-1020, chicken, 1:500) overnight at 4 °C. Sections were rinsed three times with PBS1x/0.2%TX for 5 min to remove the antibody and were incubated with the secondary antibody goat anti-chicken fluorescein labelled 488 (Aveslabs cat#F-1005, 1:1000) for 1 h in the dark. Remaining antibody was removed by rinsing two times with PBS1x for 5 min, followed by incubation with DAPI (1:5000 from 10mg mL^−1^ stock diluted in PBS1x) for 5 min. Sections were rinsed with PBS1x before mounting with DAKO fluorescence mounting medium.

### 4.6. In Situ Hybridization

#### 4.6.1. Bdnf RNA Probe Synthesis

A plasmid (pT7T3D-Pacl, 2897kb, TransOMIC Technologies, Huntsville, AL, USA) containing cDNA of BDNF (NM_007540) and promoter elements for T3 and T7 RNA polymerases was used. The plasmid was cut using restriction enzymes (HINDIII for sense, XLOI for anti-sense) and subsequently linearized. The transcription of this linearized plasmid resulted in single-strand RNA probes; the sense and the anti-sense probe. The BDNF antisense probe was used during the experiment, the sense probe served as control. The incubation with the sense riboprobe did not show any signal.

#### 4.6.2. Digoxigenin (DIG) Labelling 

Single-stranded RNA probes were labelled with DIG-nucleotides following the DIG RNA Labeling Mix procedure (Roche Diagnostics, Mannheim, Germany). The labelled RNA probes were cleaned up using the RNeasy Mini Kit (Qiagen, Hilden, Germany). Quality and quantity of the DIG-labelled probes were analyzed by electrophoresis of a 1% agarose gel with ethidium bromide staining.

#### 4.6.3. In Situ Hybridization 

For the same female brains, dorsal hippocampus was sectioned at 18 µm thickness in coronal plane and mounted on microscope slides (Superfrost Plus, Thermo Fisher Scientific, Dreieich, Germany). Slides were fixed in 4% PFA at 4 °C for 20 min and rinsed two times with PBS1x for 5 min. Slides were incubated with 0.2 M HCl for 8 min and rinsed again with PBS1x for 2 min. Treatment with 0.4 U/mL proteinase K in TE buffer (50 mM Tris/5 mM EDTA) was performed for 10 min. Slides were rinsed with PBS1x for 5 min and soaked in 4% PFA at 4 °C for 20 min. Incubation with acetic anhydride (2 × 600 µL) in 0.1 M triethanolamine was performed for 10 min followed by a wash with PBS1x for 5 min. Sections on the slides were dehydrated with saline for 5 min followed by an ethanol series of 30%, 50%, 70%, 80%, 95%, and two times 100%. Antisense (600 ng probe mL^−1^) and sense probes were dissolved in hybridization mix (50% formamide, 20 mM Tris-HCl pH 8.0, 0.3 NaCl, 5 mM EDTA pH8.0, 10% dextran sulfate, 0.02% ficol, 0.02% Polyvinylpyrrolidone, 0.02% BSA, 0.5 mg/mL tRNA (Sigma 10109517001), 0.2 mg/mL Carrier DNA, 200mM DTT) and heated for 5 min at 37 °C. Slides were incubated with the antisense or sense probe solutions overnight at 55 ± 1 °C. To prevent evaporation of the hybridization buffer, sections were covered with coverslips. After hybridization, slides were dipped in 5× saline-sodium citrate buffer (SSC) to detach the coverslips and were placed directly in 2× SSC containing 50% formamide for 30 min at 62 °C to remove remaining probe and hybridization buffer. Slides were washed with 1× SSC in 50% formamide for 30 min at 62 °C and finally with 0.1× SSC for 30 min at 62 °C to remove non-specific DNA/RNA hybridization. Slides were incubated with 4% sheep serum for 1 h. For antibody administration, slides were switched to Shandon Cassettes. Slides were rinsed two times with Tris-NaCl-Tween (TNT) buffer for 2 min at 30 °C, followed by a wash with PE buffer (Qiagen, Hilden, Germany) for 30 min at 30 °C and incubated with anti-DIG fragments (Roche Diagnostics, diluted 1:1000 in PE buffer) for 2 h at 30 °C. Slides were rinsed three times with TNT for 2 min at 30 °C and incubated with tyramine CY3 diluted (1:50) in amplification diluent (TSA Cyanine 3 Reagent Kit, Perkin Elmer, Germany) for 15 min at 30 °C in the dark. Slides were rinsed with PBS1x for 2 min and incubated with DAPI (1:5000) for 5 min. Slides were rinsed three times with PBS1x for 2 min, rinsed in autoclaved H2O and mounted using DAKO fluorescence mounting medium.

### 4.7. Microscopy 

For each brain three dorsal hippocampal sections were analyzed. Tile scans of the whole hippocampus were made using a Leica AF7000 wide field microscope equipped with a Hamamatsu-Flash4-USB3-101292 camera and LAS X software (Institute for Molecular Biology, Mainz, Germany). Images were acquired with a HC PL FLUOTAR L 20×/0.40 objective lens using the same settings for each section. Images visualizing GFP fluorescence were acquired using an L5 filter, DAPI was imaged using an A4 filter, and *Bdnf* expression (Cy3) was imaged using an N3 filter. Single images were merged to obtain a complete tile scan. Tile scans were used to enable DG and CA3 analysis in Fiji (ImageJ, Bethesda, MD, USA).

### 4.8. Intensity Quantification 

Bilateral DG and CA3 regions of the hippocampus were selected and the mean GFP and BDNF intensity were quantified using the Fiji software. The experimenter was blind to treatment conditions during quantification. Intensity levels were averaged per animal and subsequently compared between treatment groups. One ketamine-injected animal was excluded from further GFP analysis, due to experimental errors.

### 4.9. Statistical Analysis

Statistical analysis was performed with the Prism version 5 software (Graphpad software Inc., San Diego, CA, USA). All data was tested for normality with the D’Agostino and Pearson omnibus normality test. Behavioral battery data was first analyzed using a one-sample *t*-test against chance level. Battery and FST data were analyzed using a one-way ANOVA, only spontaneous alternation was analyzed using a two-way ANOVA, followed by a Bonferroni post-test. Effect of both sex and treatment on battery and FST data was evaluated using a two-way ANOVA followed by a Bonferroni post-test when applicable. Intensity values were analyzed using a one-way ANOVA followed by a Bonferroni post-test. All data are expressed as means ± SEM and were considered significant at *p* < 0.05.

## Figures and Tables

**Figure 1 ijms-21-02142-f001:**
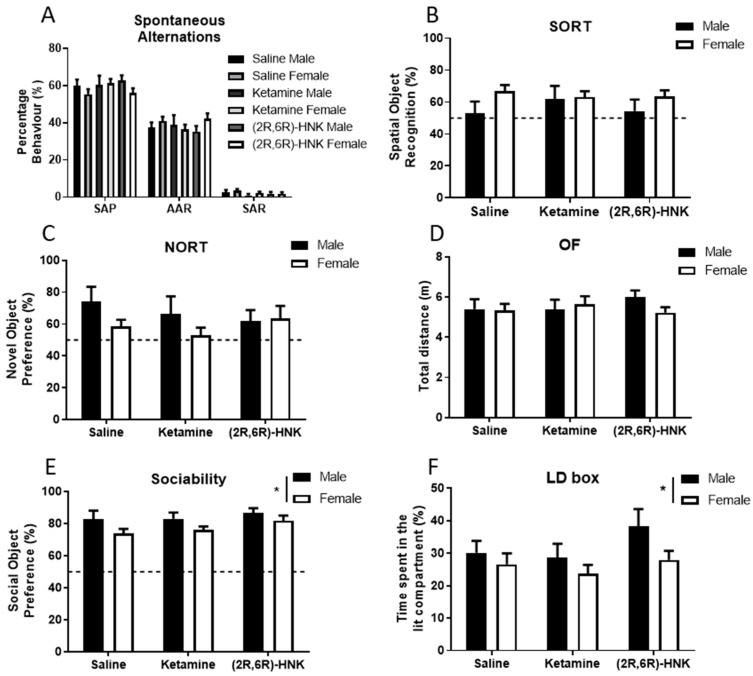
Sex, but not treatment with ketamine or (2R,6R)-hydroxynorketamine ((2R,6R)-HNK) affected sociability and anxiety of Arc-CreER^T2^ × CAG-Sun1/sfGFP mice 24 h post-injection. (**A**) Spontaneous alternations in the Y-maze. The frequency of the behaviors spontaneous alternation performance (SAP), alternate arm returns (AAR) and same arm returns (SAR) in treatment groups was counted and is presented in percentages. Male *n* = 21, female *n* = 39. (**B**) Spatial object recognition task (SORT). Exploration time of the displaced object was measured (s), spatial object recognition was calculated as and is presented in percentage. Male *n* = 21, female *n* = 39. (**C**) Novel object recognition task (NORT). Novel object interaction time was measured (s) and presented in percentage of the total exploration time. Male *n* = 16, female *n* = 30. (**D**) Locomotor activity in the open field (OF). Total distance was calculated in meters. Male *n* = 22, female *n* = 39. (**E**) Sociability test. Preference for the social object (juvenile male mouse) was measured and is presented in percentage. Sex difference was observed in the preference for the social object (*p* = 0.0133, for detailed statistics see Table 1). Male *n* = 21, female *n* = 37. (**F**) Anxiety-like behavior in the light-dark box (LD box). Time spent in the lit compartment was calculated and presented in percentage. Sex difference was observed in the time spent in the lit compartment (*p* = 0.0405, for detailed statistics see Table 1). Male *n* = 22, female *n* = 39. All behaviors are expressed as mean and error bars represent SEM, dotted line represents chance level, * *p* < 0.05.

**Figure 2 ijms-21-02142-f002:**
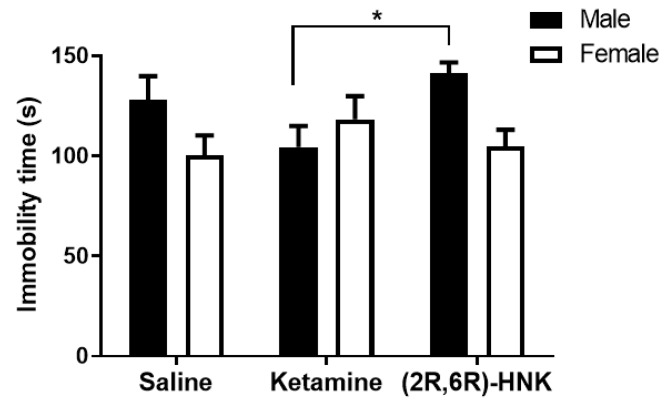
Effects on the **i**mmobility time measured in the forced swim test (FST) 24 h after a single injection with ketamine or (2R,6R)-HNK in Arc-CreER^T2^ × CAG-Sun1/sfGFP mice. (Two-way ANOVA: interaction: F(2,80) = 3.397, *p* = 0.0384; sex: F(1,80) = 3.924, *p* = 0.0510; *n* = 11–18 mice per group). Bonferroni post test showed that male mice treated with ketamine are more immobile than those treated with (2R,6R)-HNK [*t* = 2.89 df = 80, * *p* < 0.05]. Immobility time is presented in mean (s), error bars represent ± SEM.

**Figure 3 ijms-21-02142-f003:**
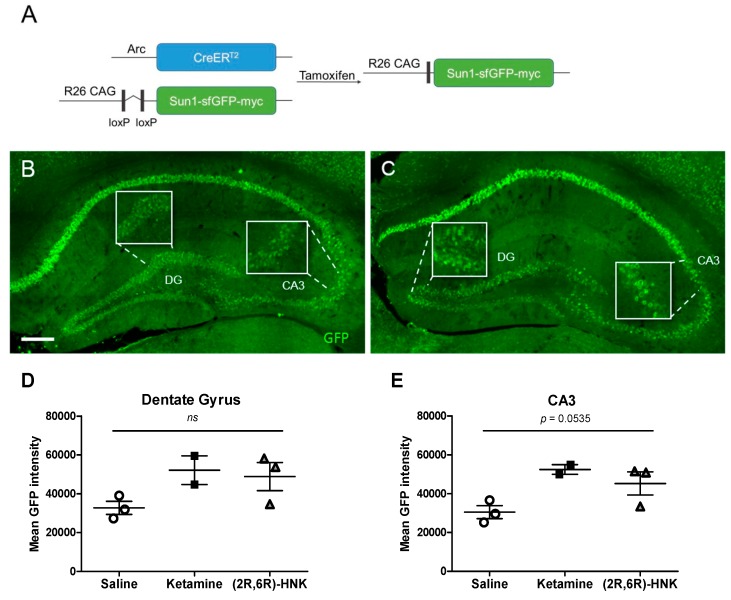
Green fluorescent protein (GFP)-labelled activated nuclei in the hippocampal dentate gyrus (DG) and Cornu Ammonis region 3 (CA3) after a single injection with ketamine or (2R,6R)-HNK in female mice. (**A**) Genetic design. Administration of tamoxifen (TM) to Arc-CreER^T2^ × CAG-Sun1/sfGFP mice induced a Cre-mediated recombination in activated CreER^T2^-expressing neurons, thus resulting in permanent GFP label on the nuclear membrane in the activated neurons. (**B**,**C**) Immunohistochemical GFP staining of coronal sections showing the DG and CA3 region of the hippocampus of female Arc-CreER^T2^ × CAG-Sun1/sfGFP mice injected with ketamine (**B**) or (2R,6R)-HNK (**C**). Magnification 20×, scale bar represents 200 µm. Insert pictures represented zoomed-in details. Ketamine and (2R,6R)-HNK did not significantly alter GFP-labelled Arc expression in DG (**D**) or CA3 (**E**). A trend towards significance is observed in CA3 (DG: one-way ANOVA, F (2, 5) = 2.965, *p* = 0.1416; CA3: one-way ANOVA, F (2, 5) = 5.566, *p* = 0.0535). Arc expression is reflected in mean GFP intensity, which is displayed in mean gray values. Error bars represent ± SEM. *N* = 2–3 mice/group, 3 sections per animal, *ns* = not significant.

**Figure 4 ijms-21-02142-f004:**
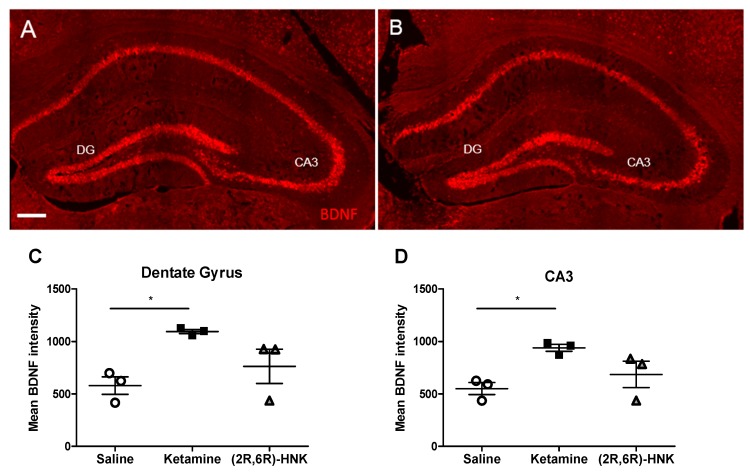
*Bdnf* mRNA expression in the hippocampal DG and CA3 after a single injection with ketamine or (2R,6R)-HNK. (**A**,**B**) In situ hybridization with *Bdnf* probes on coronal sections showing the DG and CA3 region of the hippocampus of female Arc-CreER^T2^ × CAG-Sun1/sfGFP mice injected with ketamine (**A**) or (2R,6R)-HNK (**B**). Magnification 20×, scale bar represents 200µm. Ketamine but not (2R,6R)-HNK significantly increases *Bdnf* intensity in DG (**C**) and CA3 (**D**). (DG: one-way ANOVA, F (2, 6) = 5.991, *p* = 0.0371, followed by Bonferroni post-test, ketamine vs. saline: *t* = 3.415, df = 6, *p* < 0.05. CA3: one-way ANOVA, F (2, 6) = 5.754, *p* = 0.0402, followed by Bonferroni post-test, ketamine vs. saline: *t* = 3.341, df = 6, *p* < 0.05). Mean *Bdnf* intensity is displayed in mean gray values, indicating *Bdnf* mRNA expression. Error bars represent ± SEM. *N* = 3 mice/group, 3 sections per animal. * *p* < 0.05.

**Figure 5 ijms-21-02142-f005:**
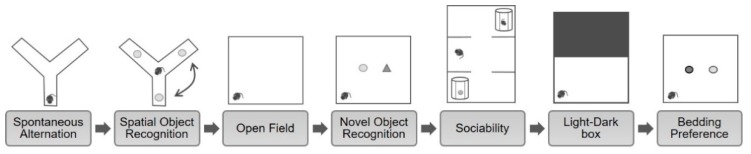
Schematic of the behavioral battery. This battery comprises a comprehensive behavioral battery to investigate multiple behavioral domains. In chronological order the battery consists of spontaneous alternation behavior in the Y-maze (working memory), spatial object recognition task (spatial memory), open field (locomotor activity), novel object recognition task (episodic memory), sociability test (social preference), light-dark box (anxiety-like behavior), and bedding preference task (olfaction) with 15 min of inter-test intervals.

**Table 1 ijms-21-02142-t001:** Statistical analysis presenting the effect of treatment and sex on the performance in different behavioral tests (two-way ANOVA). F = f-distribution, dfn = degrees of freedom numerator, dfd = degrees of freedom denominator. * Bold *p* values represent significant values.

Behavioural Test	Effect	*p* Value	F	dfn, dfd
Spontaneous alternations	Treatment	0.7364	0.3077	2, 56
	Sex	0.8434	0.0394	1, 56
	Interaction	0.0816	2.621	2, 56
SORT	Treatment	0.7892	0.2377	2, 54
	Sex	0.0678	3.475	1, 54
	Interaction	0.5034	0.6952	2, 54
NORT	Treatment	0.6674	0.4085	2, 40
	Sex	0.1274	2.423	1, 40
	Interaction	0.4187	0.8899	2, 40
OF	Treatment	0.8226	0.196	2, 55
	Sex	0.5454	0.3703	1, 55
	Interaction	0.4192	0.8833	2, 55
Sociability	Treatment	0.1729	1.816	2, 52
	**Sex**	**0.0133 ***	6.574	1, 52
	Interaction	0.8376	0.1779	2, 52
LD box	Treatment	0.1593	1.9	2, 55
	**Sex**	**0.0403 ***	4.411	1, 55
	Interaction	0.6075	0.503	2, 55

**Table 2 ijms-21-02142-t002:** Statistical analysis of the performance in the behavioral battery compared to chance level (one sample *t*-test). *t* = *t* score, df = degrees of freedom. Bold *p* values represent significant values, indicating difference compared to 50% chance level.

Behavioural Test	Treatment	Male			Female		
		*p* Value	*t*	df	*p* Value	*t*	df
SORT	Saline	0.694	0.417	5	**0.0003**	4.851	13
	Ketamine	0.2043	1.424	6	**0.0029**	3.65	13
	HNK	0.5703	0.5954	7	**0.0034**	3.717	11
NORT	Saline	0.0817	2.581	3	**0.0412**	2.311	11
	Ketamine	0.216	1.468	4	0.5445	0.6274	10
	HNK	0.149	1.655	6	0.1432	1.649	7
Sociability	Saline	**0.0016**	6.228	5	**<0.0001**	8.531	12
	Ketamine	**0.0001**	8.657	6	**<0.0001**	11.96	12
	HNK	**<0.0001**	13.02	7	**<0.0001**	9.878	11
Bedding Preference	Saline	0.0952	1.978	6	0.1845	1.402	13
	Ketamine	**0.0339**	2.736	6	0.2512	1.201	13
	HNK	**<0.0001**	11.4	7	0.3102	1.064	11

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
