# Peer review of "Sexually Dimorphic Behavioral Profile in a Transgenic Model Enabling Targeted Recombination in Active Neurons in Response to Ketamine and (2R,6R)-Hydroxynorketamine Administration"

_ijms, 2020, doi:10.3390/ijms21062142_

Round 1
Reviewer 1 Report
This was an interesting manuscript that has translational implications. Therefore, the following suggestions are made in an attempt to increase its value.
1. The route and dose of ketamine and its analog are not provided in the abstract.
2. The Introduction contains no acknowledgement that ketamine is only currently commercially available as a parenteral therapeutic, which would be a detriment to its use in chronic disorders such as depression. (Also, it induces catalepsy when used as an anesthetic in veterinary patients; the amnesia and immobilization would be considered detriments at high doses.)
3. Because the Methods are at the end of the manuscript, this reader recommends removing 'significant' from the text of paragraph 2.1.1. Stating that females had great chance above the 50% cut-off is enough. This reader recognizes that additional statistics are provided in the Appendix table. Perhaps that table should be Table 2 in the regular manuscript.
4. Some terms need to be defined when they first appear, as Methods are at the end rather than early in the manuscript. This includes OF, LD box, FST.
5. It was unclear when reading Results what determined anxiety-like behavior. The addition of a simple phrase, "...anxiety-like behavior as determined by ... was not affected...." would solve this.
6. Since Methods is at the end, rewriting the title for Figure 2 would help the reader know that Immobility times as indicated by the forced swim test were measured 24 hours after a single injection....
7. An explanation for why and when tamoxifen was used would help the reader going through paragraph 4.2 of the Methods section.
Author Response
Reviewer 1
- The route and dose of ketamine and its analog are not provided in the abstract.
Answer 1.1:
We thank Reviewer 1 for this comment. We have added this information in the abstract.
Abstract, line 22: “…We treated mice with an intraperitoneal injection of either saline, ketamine (30 mg kg-1) or (2,R,6R)-HNK (10 mg kg-1)…”
- The Introduction contains no acknowledgement that ketamine is only currently commercially available as a parenteral therapeutic, which would be a detriment to its use in chronic disorders such as depression. (Also, it induces catalepsy when used as an anesthetic in veterinary patients; the amnesia and immobilization would be considered detriments at high doses.)
We thank Reviewer 1 for raising this point.
We have added the following text:
Introduction, line 50:
“… So far, ketamine administration has had to be via parenteral administration, further impairing its applicability in the clinic. However, just recently a new ketamine nasal spray in combination with an oral antidepressant has been approved by the FDA for treatment-resistant depression, hence providing a new available formulation [7]...”
Daly et al 2019 Efficacy of Esketamine Nasal Spray Plus Oral Antidepressant Treatment for Relapse Prevention in Patients With Treatment-Resistant DepressionA Randomized Clinical Trial JAMA Psychiatry, doi: 10.1001/jamapsychiatry.2019.1189
- Because the Methods are at the end of the manuscript, this reader recommends removing 'significant' from the text of paragraph 2.1.1. Stating that females had great chance above the 50% cut-off is enough. This reader recognizes that additional statistics are provided in the Appendix table. Perhaps that table should be Table 2 in the regular manuscript.
Answer 1.3:
We thank Reviewer 2 for this suggestion we have moved Table 2 in the regular text in paragraph 2.1.1
- Some terms need to be defined when they first appear, as Methods are at the end rather than early in the manuscript. This includes OF, LD box, FST.
Answer 1.4:
We thank Reviewer 2 for this comment. We clarified those terms
Results line 139: open field (OF) and line 145: Likewise, anxiety-like behavior, measured as avoidance of the lit compartment of the light-dark (LD) box (Figure 1F) was significantly affected by sex (Table 1).
Introduction line 57 for FST “The antidepressant ketamine in mice affects several behavioral domains. It was found to be anxiolytic [8] as well as to reduce the immobility time in the forced swim test (FST), a commonly used test for depressive-like behavior”
- It was unclear when reading Results what determined anxiety-like behavior. The addition of a simple phrase, "...anxiety-like behavior as determined by .was not affected...." would solve this.
Answer 1.5:
We thank Reviewer 1 for the comment. We changed the sentence
Results, line 146 “…Likewise, anxiety-like behavior, measured as avoidance of the lit compartment of the light-dark box (LD) (Figure 1F) was significantly affected by sex (Table 1). Importantly, statistical analysis showed that anxiety-like behavior tested in the LD box was not affected by ketamine nor (2R,6R)-HNK (Table 1). Neither treatment nor sex affected olfaction in the bedding preference test (Figure A1 and Table 1)…”
- Since Methods is at the end, rewriting the title for Figure 2 would help the reader know that Immobility times as indicated by the forced swim test were measured 24 hours after a single injection....
Answer 1.6:
We thank Reviewer 1 for the comment. We changed the title
“Effects on the immobility time measured in the FST 24 hours after a single injection with ketamine or (2R,6R)-HNK in Arc-CreERT2 x CAG-Sun1/sfGFP mice”
7.An explanation for why and when tamoxifen was used would help the reader going through paragraph 4.2 of the Methods section.
Answer 1.7:
We thank Reviewer 1 for the comment. We changed paragraph 4.2
Method Section, Line 312-316:
“…To induce Cre recombination, the tamoxifen mixture was administered by a single intraperitoneal injection (150 mg kg-1) followed by a single intraperitoneal injection with one of the compounds, saline or ketamine or (2R,6R)-HNK. Tamoxifen was injected to make active CreERT2-expressing cells undergo Cre-mediated recombination, resulting in permanent expression of GFP. Intensity of the GFP protein was measured after immunohistochemistry (see paragraph 4.4 and 4.5)…”
Reviewer 2 Report
The role of ketamine-Induced behavioral changes is very interesting.
I think that potential effect on no transgenic models should be hypothized in your paper
Author Response
Reviewer 2:
Comments and Suggestions for Authors
The role of ketamine-Induced behavioral changes is very interesting.
I think that potential effect on no transgenic models should be hypothesized in your paper
Answer 2:
We thank Review 2 for this valuable point.
We did not include wild type animals in our study because we referred to the effects of ketamine described in the literature in wild type animals. The behavioral effects of ketamine on wild-type rodents are quite well studied (for example see a very recent review: Polis et al. 2019 Behavioral Brain Research, https://doi.org/10.1016/j.bbr.2019.112153). However, there is indeed no study published, analyzing the behavioral effects of HNK on wild-type mice using the same behavioral test battery employed in this study. Such an experiment would be very interesting. We highlighted these aspects within the discussion in line260:
“…Moreover, in the rodent literature there is no consensus yet on the antidepressant-like effects of ketamine. Also, these effects seem to be dependent on the dose, species and strain, test, stressor and the sex of the experimenter as highlighted in a recent review [36]…”
Reviewer 3 Report
Herzog et al. performed a baseline characterization of sex differences in Arc-CreERT2;CAG-Sun1/sfGFP mice treated with ketamine and the ketamine metabolite hydroxynorketamine ((2R, 6R)-HNK). The use of Acr-CreERT2;CAG-Sun1/sfGFP mice enables tamoxifen-dependent GFP tagging of activated neurons. The authors find that male and female Acr-CreERT2;CAG-Sun1/sfGFP mice show different response profiles in tests of sociability and anxiety that are unaffected by ketamine or by HNK treatment, while male and females respond differently in the forced swim test. They also show that ketamine increases Bdnf expression in female mice, similar to that previously reported in male mice.
No effect of ketamine or HNK was seen on immobility in the FST, which is surprising. Nor, was there an effect of ketamine or HNK or Arc expression in the hippocampus. Together these negative results raise concerns about the dosing. They did find that ketamine, but not HNK, increases BDNF expression in the hippocampus of female mice, but did not include male mice in this study.
Round 2
Reviewer 1 Report
This manuscript is now okay.
Author Response
We thank Reviewer 1 for the positive comment.
Reviewer 3 Report
The authors have provided further discussion of limitations in the study raised in the initial review.
Author Response
We thank Reviwer 2 for the comment.
We have added even more information about the limitation of the study.
Discussion line 262
"Therefore, we once more highlight that the interpretation of our findings must remain limited to the sex of the tested animals for each test/measurement and to the specific doses that we applied in our studies"